# Are sweet snacks more sensitive to price increases than sugar-sweetened beverages: analysis of British food purchase data

Richard D Smith,[1,2] Laura Cornelsen,[1] Diana Quirmbach,[1,2] Susan A Jebb,[2,3] Theresa M Marteau[2]

[1]Faculty of Public Health and Policy, London School of Hygiene and Tropical Medicine, London, UK
[2]Behaviour and Health Research Unit, Institute of Public Health, University of Cambridge, Cambridge, UK
[3]Nuffield Department of Primary Care Health Sciences, University of Oxford, Oxford, UK

**Correspondence to**
Richard D Smith;
richard.smith@lshtm.ac.uk

## ABSTRACT

**Objectives** Taxing sugar-sweetened beverages (SSBs) is now advocated, and implemented, in many countries as a measure to reduce the purchase and consumption of sugar to tackle obesity. To date, there has been little consideration of the potential impact that such a measure could have if extended to other sweet foods, such as confectionery, cakes and biscuits that contribute more sugar to the diet than SSBs. The objective of this study is to compare changes in the demand for sweet snacks and SSBs arising from potential price increases.

**Setting** Secondary data on household itemised purchases of all foods and beverages from 2012 to 2013.

**Participants** Representative sample of 32 249 households in Great Britain.

**Primary and secondary outcome measures** Change in food and beverage purchases due to changes in their own price and the price of other foods or beverages measured as price elasticity of demand for the full sample and by income groups.

**Results** Chocolate and confectionery, cakes and biscuits have similar price sensitivity as SSBs, across all income groups. Unlike the case of SSBs, price increases in these categories are also likely to prompt reductions in the purchase of other sweet snacks and SSBs, which magnify the overall impact. The effects of price increases are greatest in the low-income group.

**Conclusions** Policies that lead to increases in the price of chocolate and confectionery, cakes and biscuits may lead to additional and greater health gains than similar increases in the price of SSBs through direct reductions in the purchases of these foods and possible positive multiplier effects that reduce demand for other products. Although some uncertainty remains, the associations found in this analysis are sufficiently robust to suggest that policies—and research—concerning the use of fiscal measures should consider a broader range of products than is currently the case.

### Strengths and limitations of this study

► Detailed transaction level data on all food and beverage purchases collected electronically from a representative sample of >30 000 Great Britain households over 2 years.
► Transaction level data allow for separating and analysing demand for ready-to-consume sweet snacks.
► Demand analysis accounts for zero purchases and endogeneity of total food expenditure.
► Data exclude purchases of foods and beverages bought and consumed outside homes.
► Purchase data do not necessarily amount to consumption due to possible waste.

beverages (SSBs) due to their consistent association with energy intake, weight gain, risk of type 2 diabetes, as well as dental caries.[3] In the USA, six local jurisdictions have a tax on sugary beverages implemented due to health concerns.[4] Mexico, Finland and France apply different levels of volumetric taxes on SSBs, Hungary has adopted a system of volumetric taxes from products exceeding specified levels of sugar, and Chile taxes drinks with high levels of sugar at a rate 8% higher in comparison to drinks containing less sugar.[4] More recently, Portugal and Catalonia (Spain) implemented a two-tiered tax on sugary drinks, the United Arab Emirates and Saudi Arabia introduced a 50% tax on carbonated drinks and Brunei and Thailand introduced an excise duty on sugary drinks.[4] There are similar plans across a number of other countries such as Estonia, the Philippines, Indonesia, Israel and South Africa.[5] The UK government has confirmed an industry levy starting in April 2018 to incentivise producers to reformulate their products or, if not, to increase the price of SSBs.[6]

Research to date suggests that increasing the price of SSBs generates a small, but significant, reduction in their purchase (broadly, a

## INTRODUCTION

With the global prevalence of obesity and associated health risks continuing to increase,[1 2] health-related taxes have become an established policy option intended to reduce energy intake. Most of these have focused on sugar-sweetened

10% price rise reduces purchases by 6%–8%), with a more pronounced effect in poorer households and that substitution towards other soft drink categories only minimally offsets the energy reductions achieved through decreases in SSBs.[7–18] However, there has been little research on the impact such a price increase could have on other contributors to sugar and energy intake, including alcohol[18] and sweet snack foods (such as confectionery, cakes and biscuits). With the apparent success of fiscal measures to increase the price of SSBs, it would be useful to establish whether a similar, or possibly greater, effect on consumption of snack foods could be obtained from a similar price change.

The research presented here is the first to provide a direct analysis of the relationship between price increases and demand for sweet snack foods, within the context of demand for soft drink and alcoholic drink purchases, across different income groups.

## METHODS

The impact, or sensitivity, of demand for a product to price changes is termed the price elasticity of demand. This shows the per cent change in the demand for product X if its own price changes (own-price elasticity) or the price of other products (Y, Z) changes (cross-price elasticity). These elasticities are estimated from demand models. We apply a partial demand model, which models household expenditure shares on prices of different products and total expenditure, adjusted for overall price level. The demand model we use is adapted from the common and widely applied Almost Ideal Demand System (AIDS).

The demand model and price elasticities are estimated from household expenditure data from January 2012 to December 2013, provided by Kantar Worldpanel. The data include information on household expenditures from a sample of British households (~36 000), representative of the population with respect to household size, number of children, social class, geographical region and age group on food and drink purchases for home consumption made in a variety of outlets, including major retailers, supermarkets, butchers, greengrocers and corner shops. The dataset consists of individual transactions, providing detailed information on the day of purchase, outlet, amount spent, volume purchased and also nutrient composition of each of the products, including sugar. Households record all purchases (barcodes and the receipts) for products brought back into the home with handheld scanners at home. In addition, Kantar Worldpanel annually collects sociodemographic information for each household, such as household size and composition, income group, social class, tenure and geographical location (postcode district), as well as age, gender, ethnicity and highest educational classification of the main shopper. As we are interested in analysing the demand across income groups we excluded households (n=4075) for which this variable is missing (due to households' preference to not report this).

The full dataset used in the analysis thus consists of 32 249 households, of which 80% appear in both years (25 535), providing ~75 million food and beverage purchases disaggregated at the brand and package level, capturing both cross-sectional and longitudinal variation in household purchases.

For analysis, data were aggregated from all foods and beverages into 13 distinct groups: (1) high-sugar soft drinks, containing more than 8 g sugar/100 mL (assuming a dilution rate of 1:4 as used by the British Soft Drinks Association for concentrated SSBs); (2) medium-sugar soft drinks, with between 5 and 8 g sugar/100 mL; (3) low-sugar soft drinks with less than 5 g of sugar/100 mL; (4) other soft drinks, including fruit juices, milk-based drinks (excluding pure milk) and water[i]; (5) alcohol, including beer, lager, cider, wines and spirits; (6) cookies, biscuits and cereal bars; (7) chocolate and confectionery; (8) cake-type snacks, including cake bars, pastries, muffins, flapjack and mince pies; (9) savoury snacks, including crisps, popcorn, crackers and savoury assortments; (10) fresh and frozen meat and fish; (11) dairy; (12) fruit and vegetables; (13) rest of food and drink. Sweet snack foods—defined as foods which are at ambient temperature and able to be consumed on the go without utensils—were the most disaggregated as these were the focus for this study.

As many beverages and snack foods are storable and not purchased very frequently, data were aggregated at 4-week intervals for each household, providing a total of n=6 23 459 household-month observations. As the data are aggregated to 4-weekly periods (n=26) and into 13 groups, we estimate geographical price indices from transaction prices of each individual product, based on the postcode area the households reside (see online supplementary appendix 1 for further details).

Even at this level of aggregation, a substantial amount of zero-expenditure months remain, as most households do not buy beverages or foods from every category every month and some households never buy certain categories during the whole sample period. A two-step procedure was followed to take account of this censoring of the dependent variable in the estimation strategy. The AIDS approach was adapted for the panel data context to allow control for unobserved household heterogeneity via a fixed-effects specification. The full specification, including the procedures for handling censoring, endogeneity of prices and total expenditure and estimation of price elasticities is provided in online supplementary appendix 1.

---

[i] The categorisation of the non-alcoholic beverages follows the structure in the proposed levy for sugary drinks producers in the UK (effective April 2018),[6] separating drinks that would be levied at higher rate of £0.24 per litre (drinks containing more than 8 g of sugar per 100 mL), at a lower rate of £0.18 per litre (drinks containing between 5–8 g of sugar/100 mL) and not levied (drinks <5 g sugar/100 mL) and remaining soft drinks (juice with no added sugars, milk-based drinks and water).

**Table 1** Demographic characteristics of estimation sample

| | All households | Low income | Middle income | High income |
|---|---|---|---|---|
| Number of households | 32 249 | 11 580 | 15 816 | 4853 |
| Number of observations | 623 459 | 223 174 | 305 841 | 94 444 |
| Household size (SD) | 2.7 (1.3) | 2.3 (1.3) | 2.9 (1.3) | 3.2 (1.2) |
| Age of main shopper (SD) | 47.8 (15.3) | 52.4 (17.0) | 46.0 (14.3) | 42.9 (10.8) |
| Number of children if have children (SD) | 1.7 (0.8) | 1.8 (0.9) | 1.8 (0.9) | 1.7 (0.8) |
| Share of households that have children | 0.4 (0.5) | 0.3 (0.5) | 0.4 (0.5) | 0.5 (0.5) |
| Social grade (%) | | | | |
| Class A and B (highly skilled) | 20.2 | 5.7 | 20.9 | 52.5 |
| Class C1 | 37.5 | 30.5 | 43.0 | 36.2 |
| Class C2 | 18.0 | 15.6 | 22.4 | 9.2 |
| Class D | 13.9 | 22.0 | 11.7 | 1.7 |
| Class E (unskilled) | 10.4 | 26.2 | 1.9 | 0.3 |
| Highest qualification (%) | | | | |
| Degree or higher | 24.1 | 11.6 | 25.9 | 47.8 |
| Higher education | 13.5 | 11.6 | 15.2 | 12.1 |
| A level | 11.6 | 10.0 | 13.2 | 10.6 |
| Secondary education (GCSE) | 18.8 | 22.2 | 18.8 | 10.8 |
| Other | 7.6 | 11.6 | 6.0 | 3.1 |
| None | 7.6 | 15.2 | 4.1 | 0.9 |
| Unknown | 16.8 | 17.9 | 16.7 | 14.6 |
| Tenure (%) | | | | |
| Owned outright | 24.2 | 29.5 | 22.8 | 16.2 |
| Mortgaged | 40.0 | 17.1 | 47.6 | 69.7 |
| Rented | 29.7 | 46.4 | 23.6 | 9.8 |
| Other | 1.5 | 1.8 | 1.4 | 0.8 |
| Unknown | 4.7 | 5.2 | 4.7 | 3.6 |

High income >£50 000+; middle income £20 000–£49 000; low income <£20 000 per year.
GCSE, General Certificate of Secondary Education.

Due to potential differences in purchasing behaviour, the analyses are carried out in the full sample and in subsamples by household annual income (low income (<£20 000), middle income (£20 000–£49 000) and high income (>£50 000+).

## RESULTS

Table 1 presents the sociodemographic profile of the sample. A comparison of Kantar Worldpanel with representative household data from the Living Cost and Food survey (LCF)[ii] has found the sociodemographic and regional profiles of the samples to match well, although our sample has a slightly higher share of (1) low-income households, (2) households that own a computer and/or a car and (3) households in the South and Southeast of England.[19]

Table 2 (top panel) presents the average sugar content across the food and beverage groups as well as total purchases of sugar (expressed as grams per person per day) that are purchased and brought home (ie, excluding purchases consumed outside homes), across each of the categories outlined above and split by income level. There is a clear income gradient: those on lower incomes purchase more sugar per person per day. It is also clear that more sugar is consumed across all income groups from sweet snacks (17.1 g) than all beverages combined (alcoholic and non-alcoholic) (13.9 g). In comparison to SSBs in particular (6.9 g), sweet snacks combined contribute more than twice the amount of sugar. It is also evident that sweet snacks have per 100 g a considerably higher sugar content in comparison to 100 mL of beverages.

The bottom panel of table 2 shows the share of households that purchase products from each of the food groups during the 26 4-week periods. A higher share of non-purchases (eg, only 13% of households purchase medium-sugar soft drinks across the periods) has implications

---

[ii]LCF is a survey of household spending and the cost of living in the UK reflecting household budgets and is conducted by the UK Office for National Statistics.

**Table 2** Purchases of sugar (g) per person and day in 2013 and share (%) of non-zero observations across the food groups

| Food group | Average sugar content* g (SD) | All households | Low income | Middle income | High income |
|---|---|---|---|---|---|
| | | Total sugar purchased per day per person (g)† | | | |
| SSB | | | | | |
| High-sugar soft drinks | 10.4 (1.7) | 6.3 | 7.6 | 6.8 | 4.5 |
| Medium-sugar soft drinks | 6.5 (0.8) | 0.6 | 0.7 | 0.6 | 0.4 |
| Low-sugar soft drinks | 1.0 (1.4) | 1.1 | 1.2 | 1.2 | 0.9 |
| Other soft drinks (including milk based) | 7.5 (4.7) | 3.9 | 3.8 | 4.2 | 4.0 |
| Alcohol | 1.4 (1.9) | 2.0 | 2.2 | 2.3 | 1.6 |
| Sweet snacks | | | | | |
| Biscuits and cookies (including cereal fruit bars) | 29.8 (10.5) | 7.1 | 8.8 | 7.3 | 4.6 |
| Chocolate and confectionery | 48.7 (11.9) | 7.7 | 9.9 | 7.7 | 5.2 |
| Cake-type snacks | 19.9 (11.4) | 2.3 | 2.8 | 2.2 | 1.5 |
| Savoury snacks | 5.2 (8.1) | 0.6 | 0.7 | 0.6 | 0.5 |
| Fresh and frozen unprocessed meat, fish | 1.0 (1.8) | 0.5 | 0.6 | 0.6 | 0.4 |
| Dairy and eggs | 4.2 (5.0) | 15.7 | 19.6 | 15.9 | 11.4 |
| Fruit and vegetables | 6.2 (7.3) | 17.6 | 20.7 | 17.9 | 14.2 |
| Rest food and drink | 13.2 (19.2) | 57.8 | 74.2 | 57.4 | 39.4 |
| Total | | 123.2 | 152.8 | 124.6 | 88.5 |

| Food group | % of households that purchased products across the 4-week periods (non-zero observations) | | | |
|---|---|---|---|---|
| SSB | | | | |
| High-sugar soft drinks | 49 | 45 | 51 | 48 |
| Medium-sugar soft drinks | 13 | 13 | 14 | 14 |
| Low-sugar soft drinks | 69 | 64 | 72 | 72 |
| Other soft drinks (including milk based) | 55 | 47 | 58 | 65 |
| Alcohol | 51 | 43 | 54 | 59 |
| Sweet snacks | | | | |
| Biscuits and cookies (including cereal fruit bars) | 77 | 76 | 78 | 74 |
| Chocolate and confectionery | 69 | 69 | 70 | 67 |
| Cake-type snacks | 37 | 37 | 38 | 35 |
| Savoury snacks | 80 | 75 | 82 | 82 |
| Fresh and frozen unprocessed meat, fish | 91 | 89 | 92 | 92 |
| Dairy and eggs | 99 | 99 | 99 | 99 |
| Fruit and vegetables | 97 | 96 | 98 | 98 |
| Rest food and drink | 99 | 99 | 99 | 99 |

High-sugar soft drinks: >8 g of sugar/100 mL; low-sugar soft drinks: <5 g of sugar/100 mL; medium-sugar soft drinks: 5–8 g of sugar/100 mL; other soft drinks: water, fruit juice with no added sugars and milk-based drinks.

*Average sugar content per 100 g/100 mL or item/unit (cake-type snacks and chocolate and confectionery) as reported in data.

†Sugar purchases per person across the food groups are based on full dataset of 2013 only (n=32 620), aggregated first to total GB using weights provided by Kantar Worldpanel and divided by number of persons (total GB and by income groups) and days in a year. Total GB population figures are based on Kantar Worldpanel estimates of the number of households in income brackets, taking into account the share of households of different sizes (one, two, three or four members and for households that had five or more members we used an average size of five). Total Great Britain population estimate (2013): ~59.5 million, from which 27% are in households with annual income <£20 000 (low income), 40% are in households with income £20 000–£49 000 (middle income) and 17% are in households with income >£50 000 (high income). Households for which income is unknown or unanswered are excluded (14%).

SSB, sugar-sweetened beverages.

for methodology which are discussed in appendix but also provides an overview of the regularity of purchases. Approximately half of the households (49%) purchase high-sugar soft drinks across the 26 4-week periods. Low-sugar soft drinks are bought more frequently (69% of observations are positive across household periods). In comparison, cookies and biscuits as well as chocolate and confectionery are bought more frequently (77% and 69%) and cake-type snacks are bought less frequently (37%). In comparison to low and high-income households, middle-income households have a slightly higher frequency of purchase of high-sugar soft drinks and sweet snacks.

Table 3 presents total expenditure, expenditure shares and average prices across all households and split into three income groups. The critical aspect for analysis here is the expenditure share, where there is a marked income gradient with respect to expenditure on beverages and a slightly lower gradient for sweet snacks. The low-income group spend 14% of total drink expenditure on the high and medium-sugar soft drinks, compared with 12% and 10% for medium and high-income groups, respectively. Similarly, of the total food expenditure, sweet snacks represent 7%, 7% and 6% among the low, medium and high-income groups, respectively.

The full results of the unconditional, uncompensated own-price and cross-price elasticities are presented in online supplementary appendix 2. In sum, the own-price elasticity for alcoholic drinks is higher than for all other categories; that is, alcoholic drinks are more sensitive to price change than any other category. Elasticities for all categories are inelastic (ie, smaller than 1); this means that there is a less than proportionate decrease in purchase following a price rise for products, indicating that price increases reduce demand for all products, although with differing strength of effect. This pattern is seen across all income groups, with relatively similar absolute elasticity values. Comparing SSB and sweet snack price sensitivity, the elasticity for SSB is on average −0.77 (a 10% increase in price yields a 7.7% reduction in quantity purchased), whereas for chocolate and confectionery it is −0.74, biscuits −0.69 and cakes −0.66. There is relatively little variance across income groups in the own-price elasticity for chocolate and confectionery, whereas for biscuits and cookies and cake-type snacks, low-income households are relatively more price responsive (−0.74 and −0.71, respectively, in comparison to −0.64 and −0.53 in high-income group). Sweet snack foods, overall, thus appear to have only slightly lower level of price sensitivity in comparison to SSBs.

Of interest also is the impact on purchases across other aspects of the diet when the price of SSBs or sweet snacks increases. Figures 1-4 present the impacts on purchases as a result of a 1% increase in price of each of the soft drink and snack categories to illustrate the variance in these effects (presenting only those effects where CIs exclude zero). This is presented for the total sample (figure 1) and then for each income group (figures 2-4).

In aggregate across all income groups, (figure 1) clear differences arise from increasing the price of SSBs compared with sweet snacks. Increases in the price of high-sugar soft drinks are associated with a decrease in purchases of medium-sugar soft drinks (2.5% reduction in purchase if the price of high-sugar drinks increases by 10%) but increased purchases of other soft drinks (1.1%) and chocolate and confectionery (0.08%). Increasing the price of diet/low-sugar drinks elicits greater reaction in other soft drink purchases (1.1% decrease in purchase of high-sugar drinks and 2.8% decrease in purchase of medium-sugar drinks for a 10% increase in price of low-sugar drinks), but also some increase in demand for cakes, biscuits and chocolate (1.3%–1.7%). Increasing the price of medium-sugar soft drinks, however, only reduces demand for other soft drinks (by 0.5%), low-sugar soft drinks (0.3%) and alcohol (0.3%) with no associations observed with demand for snacks.

For sweet snacks, there are considerably more complementary effects, with significant reductions in other categories. A price increase for chocolate and confectionery items is associated with small but significant decreases across all soft drinks (reductions in purchase of 0.6%–0.8% for a 10% price increase) as well as biscuits and cakes (by 1.2%) and savoury snacks (1.6%). For biscuits, there are significant reductions in the demand for cakes (2.3%) as well as chocolate and confectionery (1.7%). Finally, for a price increase in cakes, there are smaller changes, with reductions in purchases of biscuits (by 0.7%), but increases in the purchase of chocolate and confectionery (0.7%) and alcohol (0.8%). Thus, increasing the price of chocolate snacks especially elicits a range of significant reductions in purchases across most categories.

Although many of the associations at the aggregate level are replicated across income groups (figures 2-4), there is some clear variance by income group. An increase in the price of sugary drinks is associated with a reduction in medium-sugar drinks only within the low-income group (by 3% if price increases by 10%) while an increase in other soft drinks is observed in medium and high-income groups (1%). Furthermore, in the high-income group, a higher SSB price leads to an increase in purchases of chocolate and confectionery (1%–2%) but also a reduction in purchases of cake-type snacks (2%, although all with relatively large CIs).

Increasing the price of diet/low-sugar drinks seems to be associated with more substitute relationships, with significant increases in sweet snack demand (1%–2% increase to a price increase of 10%), especially for low and medium-income groups. However, for increases in the price of sweet snacks the differences are more marked. Increasing the price of biscuits generates complementary reductions in the purchase of chocolate and confectionery for the low-income group (by 3% if price increases by 10%), reductions in cake-type snacks for the middle-income group (3%) but no such reductions for the high-income group where a reduction in medium-sugar drinks is observed instead (8%). While a relatively large change,

**Table 3**  Mean total expenditure, expenditure shares and prices

| | All households (n=6 23 459) | | Low income (n=2 23 174) | | Middle income (n=3 05 841) | | High income (n=94 444) | |
|---|---|---|---|---|---|---|---|---|
| | Mean | SD | Mean | SD | Mean | SD | Mean | SD |
| Total 4-weekly expenditure (£) | 183.5 | 110.6 | 155.0 | 96.3 | 194.1 | 112.2 | 211.9 | 121.3 |
| Expenditure share | | | | | | | | |
| SSB | | | | | | | | |
| High-sugar soft drinks | 0.015 | 0.028 | 0.015 | 0.032 | 0.015 | 0.027 | 0.013 | 0.015 |
| Medium-sugar soft drinks | 0.002 | 0.008 | 0.002 | 0.009 | 0.002 | 0.008 | 0.002 | 0.002 |
| Low-sugar soft drinks | 0.023 | 0.033 | 0.022 | 0.033 | 0.024 | 0.032 | 0.026 | 0.023 |
| Other soft drinks | 0.016 | 0.026 | 0.013 | 0.025 | 0.016 | 0.025 | 0.020 | 0.028 |
| Alcohol | 0.079 | 0.125 | 0.071 | 0.127 | 0.083 | 0.126 | 0.087 | 0.124 |
| Sweet snacks | | | | | | | | |
| Biscuits and cookies (including cereal fruit bars) | 0.025 | 0.029 | 0.026 | 0.031 | 0.025 | 0.028 | 0.022 | 0.026 |
| Chocolate and confectionery | 0.028 | 0.041 | 0.031 | 0.045 | 0.027 | 0.038 | 0.024 | 0.037 |
| Cake-type snacks | 0.006 | 0.012 | 0.007 | 0.014 | 0.006 | 0.011 | 0.005 | 0.010 |
| Savoury snacks | 0.029 | 0.030 | 0.028 | 0.032 | 0.029 | 0.030 | 0.028 | 0.028 |
| Fresh and frozen unprocessed meat, fish | 0.129 | 0.092 | 0.122 | 0.095 | 0.130 | 0.090 | 0.137 | 0.092 |
| Dairy and eggs | 0.131 | 0.068 | 0.136 | 0.073 | 0.129 | 0.065 | 0.125 | 0.063 |
| Fruit and vegetables | 0.130 | 0.088 | 0.124 | 0.090 | 0.129 | 0.085 | 0.142 | 0.088 |
| Rest food and drink | 0.389 | 0.120 | 0.403 | 0.127 | 0.385 | 0.116 | 0.370 | 0.114 |
| All drinks | 0.134 | | 0.123 | | 0.140 | | 0.147 | |
| All food | 0.866 | | 0.877 | | 0.860 | | 0.853 | |
| % of drinks expenditure spent on SSB | 12% | | 14% | | 12% | | 10% | |
| % of food expenditure spent on sweet snacks | 7% | | 7% | | 7% | | 6% | |
| Price per volume unit (L, kg)* | | | | | | | | |
| SSB | | | | | | | | |
| High-sugar soft drinks | 0.92 | 0.74 | 0.91 | 1.06 | 0.92 | 1.06 | 0.93 | 1.07 |
| Medium-sugar soft drinks | 0.95 | 0.49 | 0.95 | 1.17 | 0.95 | 1.18 | 0.97 | 1.18 |
| Low-sugar soft drinks | 0.69 | 0.50 | 0.69 | 1.10 | 0.69 | 1.10 | 0.71 | 1.11 |
| Other soft drinks | 0.86 | 1.08 | 0.86 | 1.08 | 0.86 | 1.08 | 0.87 | 1.08 |
| Alcohol | 4.67 | 1.13 | 4.65 | 1.13 | 4.67 | 1.13 | 4.75 | 1.13 |
| Sweet snacks | | | | | | | | |
| Biscuits and cookies (including cereal fruit bars) | 3.77 | 1.07 | 3.76 | 1.06 | 3.77 | 1.07 | 3.80 | 1.07 |
| Chocolate and confectionery | 0.77 | 1.33 | 0.77 | 1.33 | 0.77 | 1.33 | 0.78 | 1.33 |
| Cake-type snacks | 1.00 | 1.06 | 0.99 | 1.06 | 1.00 | 1.06 | 1.00 | 1.06 |
| Savoury snacks | 6.46 | 5.39 | 6.44 | 1.04 | 6.46 | 1.04 | 6.51 | 1.05 |
| Fresh and frozen unprocessed meat, fish | 5.65 | 4.62 | 5.62 | 1.06 | 5.65 | 1.06 | 5.71 | 1.07 |
| Dairy and eggs | 0.98 | 0.78 | 0.98 | 1.07 | 0.98 | 1.07 | 0.99 | 1.07 |
| Fruit and vegetables | 1.66 | 1.30 | 1.65 | 1.09 | 1.66 | 1.09 | 1.69 | 1.10 |
| Rest food and drink | 2.26 | 1.91 | 2.25 | 1.05 | 2.26 | 1.06 | 2.29 | 1.06 |

*Average unit prices (£) over geographical areas (n=110); volume of cakes and chocolate and confectionery is measured by items; low income <£20 000 per year; middle income £20 000–£49 000; high income >£50 000+; high-sugar soft drinks: >8g of sugar/100 mL; medium-sugar soft drinks: 5–8g of sugar/100 mL; low-sugar soft drinks: <5g of sugar/100 mL; other soft drinks: water, fruit juice with no added sugars and milk-based drinks.
SSB, sugar sweetened beverages.

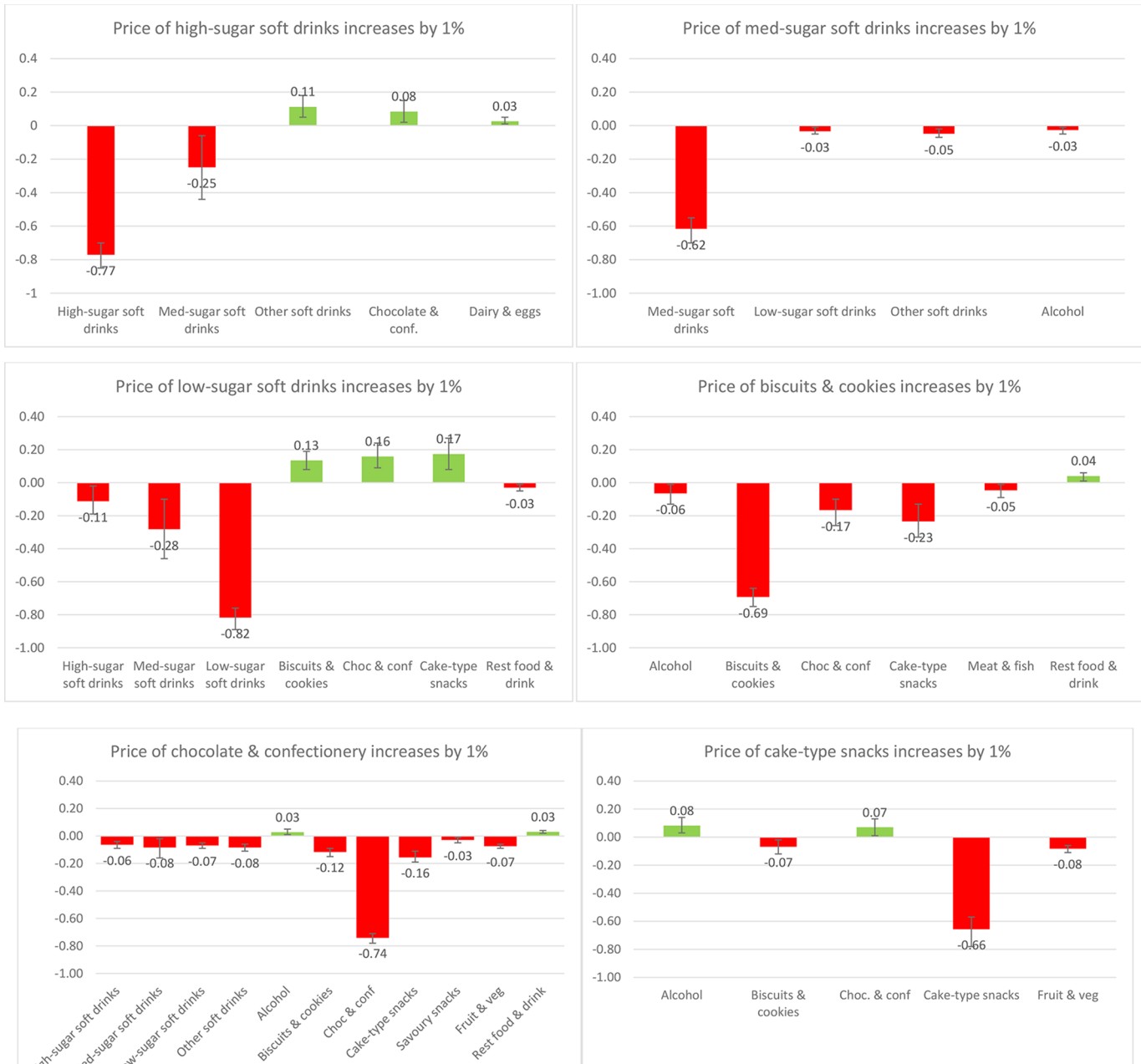

Notes: High-sugar soft drinks: >8g of sugar/100ml; medium-sugar soft drinks: 5-8g of sugar/100ml; low-sugar soft drinks: <5g of sugar/100ml; other soft drinks: water, fruit juice with no added sugars and milk-based drinks; 95% bias-corrected confidence intervals based on n=250 bootstrap replications.
Figures show elasticities for which the 95% CI excluded zero. For full set of elasticity, estimates see appendix 2.

**Figure 1** Change in demand (%) as a response to 1% price increase in soft drinks and sweet snacks (all households n=623 459).

the absolute change would be small as the share of medium-sugar drinks in overall expenditure is very small.

Changes in the price of cake-type snacks has limited impact on other categories for those in the low-income group, but for the middle-income group it reduces purchase of biscuits (1%), but is also associated with a slight increase in purchase of alcohol (1%). For the high-income group this effect is even more pronounced, with increases in purchase of alcohol (1%) and chocolate as substitutes (3%). Increasing the price of chocolate and confectionery has a similar effect across all income groups, with associated reductions in the

purchase of most other food and drink categories (1%–2% if price increases by 10%).

## DISCUSSION

The price elasticity of chocolate and confectionery was highest among the sweet snacks and is almost identical to that for SSBs (although both are lower than alcohol). Further, price increases in SSBs are associated with an increase in purchase of other soft drinks and chocolate and confectionery, whereas an increase in the price of chocolate

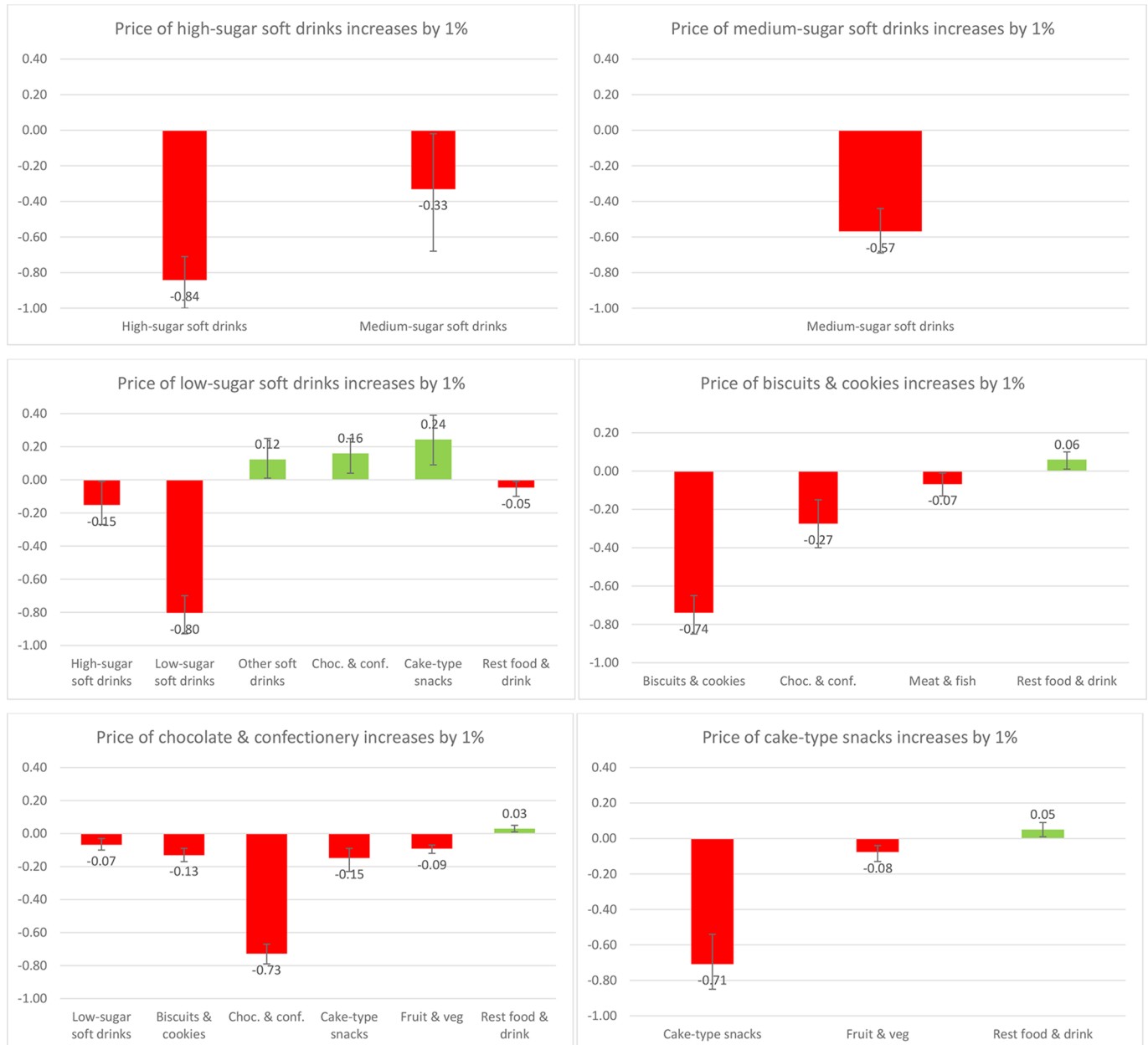

Notes: High-sugar soft drinks: >8g of sugar/100ml; medium-sugar soft drinks: 5-8g of sugar/100ml; low-sugar soft drinks: <5g of sugar/100ml; other soft drinks: water, fruit juice w no added sugars and milk-based drinks; 95% bias-corrected confidence intervals based on n=250 bootstrap replications.
Figures show elasticities for which the 95% CI excluded zero. For full set of elasticity, estimates see appendix 2.

**Figure 2** Change in demand (%) as a response to 1% price increase in soft drinks and sweet snacks (low-income households n=223 174).

is associated with a reduction in purchase of SSBs, as well as a range of other snacks. The differences across food categories and income groups indicate the complexity of estimating the impact of a single price increase. Nonetheless, it does suggest that policies to increase the price of sweet snacks could have a greater impact than that seen thus far for SSBs, not least because chocolate and confectionery alone contribute a similar quantity of sugar per person per day as SSBs in our sample. Moreover this analysis suggests they have stronger associations with reductions in other categories of foods and SSBs (ie, complementary relationships), creating a cumulative positive multiplier effect. This appears to be

most pronounced in the low and middle-income groups, as would be expected. The strength of these results suggests that further research is warranted to analyse the impact on diet composition and model the long-term impacts of such interventions on health outcomes.

The extent to which a levy on sugary snacks could yield a lower consumption of sugar is, of course, dependent on the structure of the levy, but considering the relatively high sugar content of these foods (per 100 g) even a small levy based on sugar content is likely to change prices, assuming it is passed through. Whether a multitiered levy based on sugar content, such as proposed for the sugary drinks, would encourage

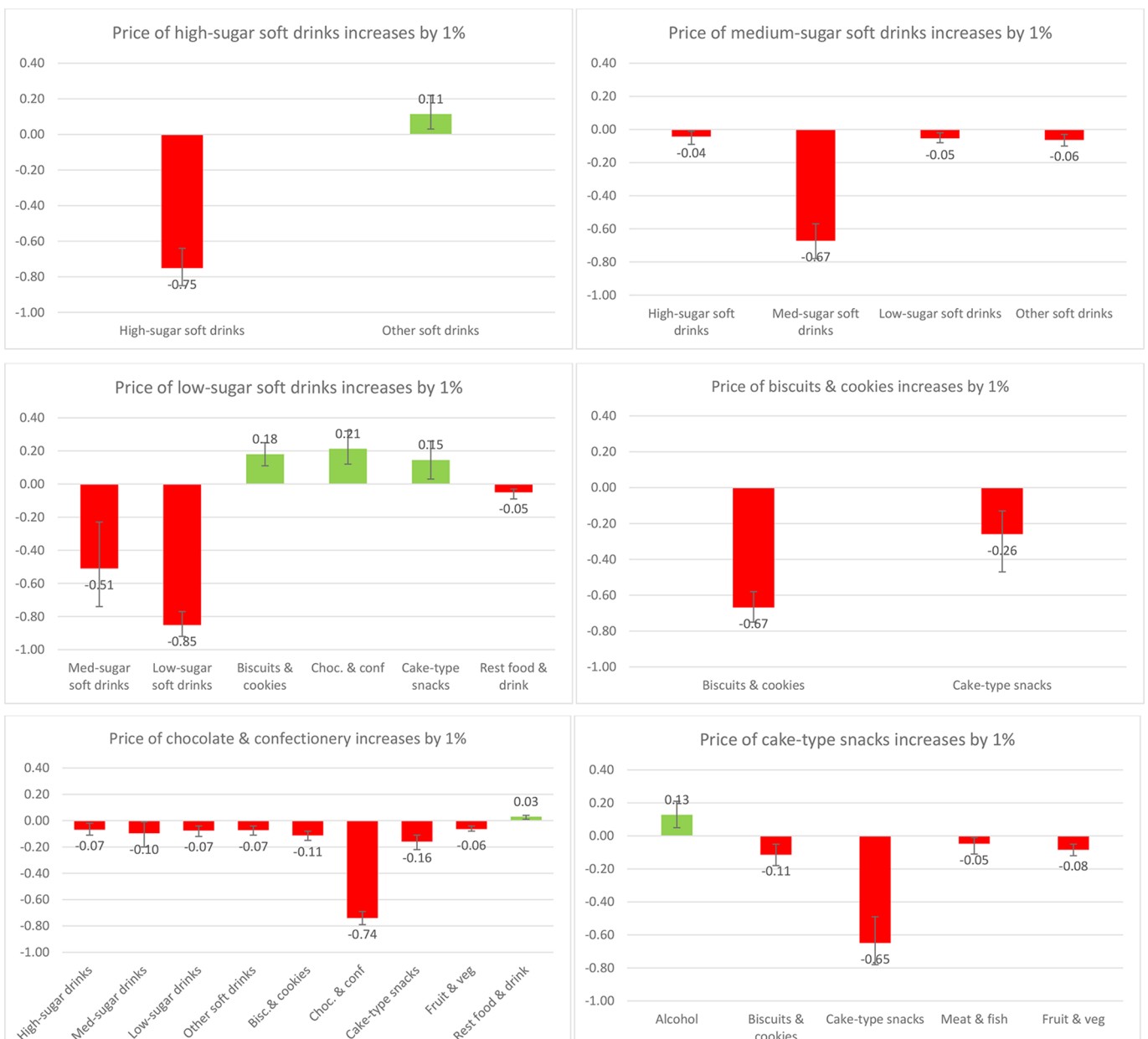

Notes: High-sugar soft drinks: >8g of sugar/100ml; medium-sugar soft drinks: 5-8g of sugar/100ml; low-sugar soft drinks: <5g of sugar/100ml; other soft drinks: water, fruit juice w no added sugars and milk-based drinks; 95% bias-corrected confidence intervals based on n=250 bootstrap replications.
Figures show elasticities for which the 95% CI excluded zero. For full set of elasticity, estimates see appendix 2.

**Figure 3** Change in demand (%) as a response to 1% price increase in soft drinks and sweet snacks (mid-income households n=305 841).

reformulation is another question since there are important differences in the ease of reformulation compared with SSBs and less is known about consumer acceptability of the reformulated snack food products.

Overall, our estimates of price elasticity for foods and sugary beverages are consistent with the literature. Meta-analyses of price elasticity in broad food groups in high-income countries find these to range between −0.4 to −0.8 and that of sweets, confectionery and sweetened beverages at −0.6.[7 20] Our estimates range between −0.6 and −0.8 but we also use greater disaggregation of food and beverage groups. Another study reports the metaestimate of price elasticity of SSBs to be −1.3 that is higher than our estimate of −0.77;

however, the metaestimate includes studies from Mexico and Brazil and price elasticity is dependent on income levels and lower income populations are likely to have greater responsiveness to price changes (ie, smaller elasticity value) as they spend a greater proportion of their incomes on food and beverages.[21] Two studies from Chile also suggest somewhat more responsive demand (SSBs: −1.3 to −1.4, sweets and desserts −0.8 to −1.2).[22 23] Elsewhere, a US study found, as here, a substitution effect towards juice and milk and a reduction in diet beverages if the price of SSBs increases. This study also estimated price elasticity for SSBs at −0.8 and a somewhat less price responsive demand for sweets and sugars than our analysis (−0.3).[24] It has to be noted however,

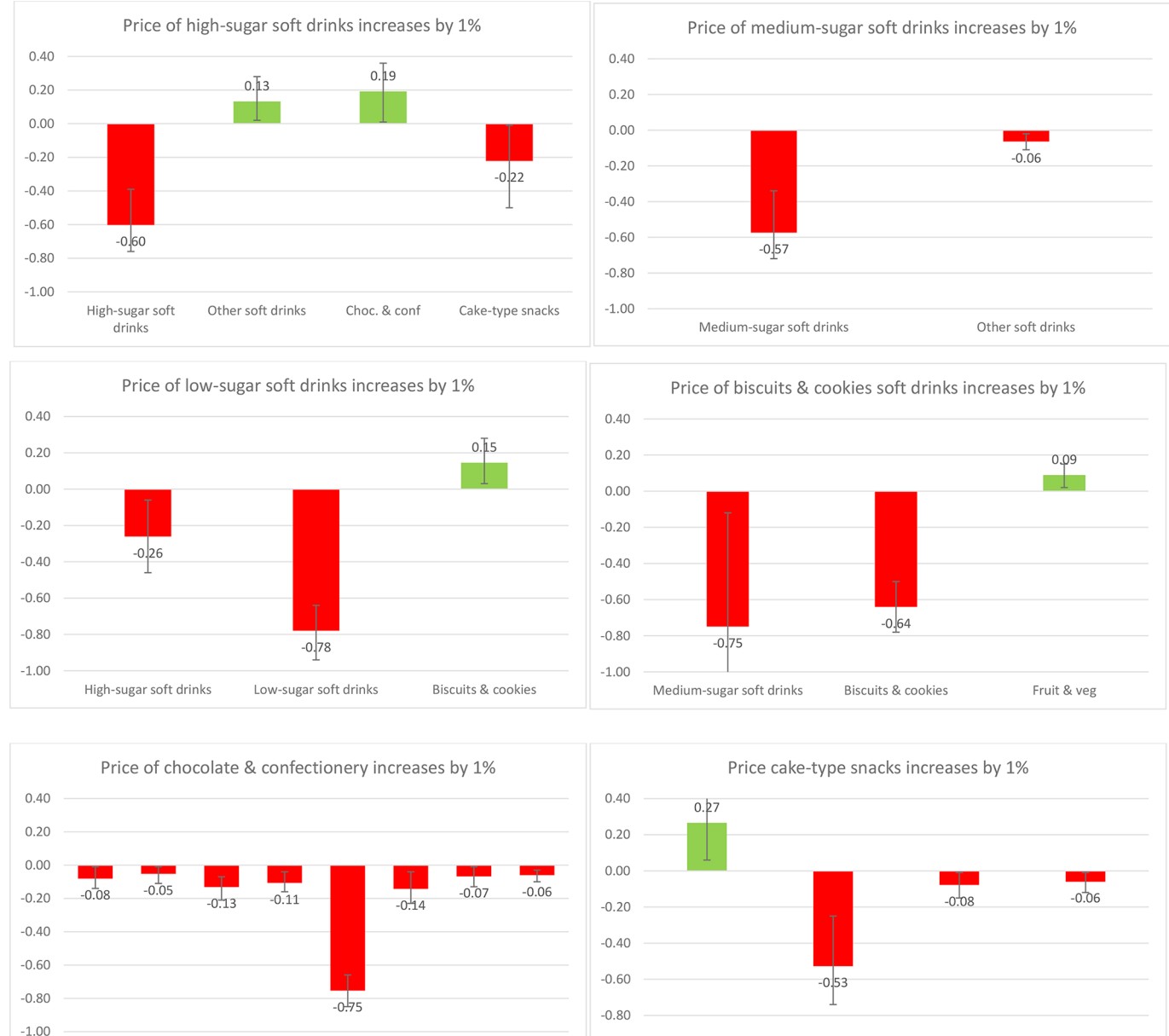

Notes: High-sugar soft drinks: >8g of sugar/100ml; medium-sugar soft drinks: 5-8g of sugar/100ml; low-sugar soft drinks: <5g of sugar/100ml; other soft drinks: water, fruit juice w no added sugars and milk-based drinks; 95% bias-corrected confidence intervals based on n=250 bootstrap replications.
Figures show elasticities for which the 95% CI excluded zero. For full set of elasticity, estimates see appendix 2.

**Figure 4**  Change in demand (%) as a response to 1% price increase in soft drinks and sweet snacks (high-income households n=94 444).

that we cannot impose a priori expectations for underlying preferences for foods and beverages to be the same in different populations and over time so some variance in elasticity estimates would be natural even if methods applied by the studies are similar.

There are, of course, limitations to the analysis presented here. The data, although large, representative and detailed, may be subject to under-recording; an issue present in all types of survey data. For instance, Kantar Worldpanel data appear to have lower levels of recorded alcohol expenditure than the Living Cost and Food survey.[19] The data also include foods and beverages purchased and brought home

and thus exclude all purchases that are consumed outside the home which are likely to be higher among more affluent households. Furthermore, the price responsiveness is based on price variations occurring in the market. This implies that any likely effect of the taxes inferred from these elasticities is subject to bias if the taxes, when implemented, have an impact on the demand beyond the direct price change.

Regardless of the models used, estimating demand requires a number of assumptions (see online supplementary appendix 1), which may have influenced the estimates. We prioritised an approach that allowed controlling for unobservable household heterogeneity, including in the

preferences towards different types of drinks and snacks while also adjusting for non-purchase and endogeneity issues. Overall, own-price elasticities are estimated with greater robustness as an a priori expectation of an inverse relationship with price exists and own-price changes have a noticeable impact on purchases. However, the estimation of cross-price elasticities (substitution or complementarity effects) across products are harder to capture, as these are generally much smaller and the direction cannot be assumed a priori.[25] As most of cross-price elasticities are estimated close to zero, even small changes in methods can possibly affect the direction and thus interpretation of the effect. In addition, price elasticities are interpreted individually (ie, allowing one price change at a time) but categories defined in this study might be taxed simultaneously (eg, high and medium-sugar soft drinks) which means that the policy impact may vary. Perhaps more critically, although this analysis can highlight significant relationships between products purchased, it cannot explain why these relationships exist. This requires further primary research and research within population subgroups.

## CONCLUSION

Increasing the price of SSBs has become an accepted policy to reduce sugar intake. Analysis presented here based on data from Great Britain suggests that extending fiscal policies to include sweet snacks could lead to larger public health benefits, both directly by reducing purchasing and therefore consumption of these foods, and indirectly by reducing demand for other snack foods and indeed SSBs. Although some uncertainty remains, the associations observed in this analysis are sufficiently robust to suggest that policies—and research—concerning the use of fiscal measures to reduce intake of free sugars and improve diet quality should consider extending beyond SSBs to include the more frequently consumed sugar-based snacks including cakes, biscuits and, especially, chocolate and confectionery.

**Contributors** RDS and TM conceived the study. DQ and LC conducted analyses, interpreted results and drafted the paper. RDS, TM and SAJ helped design the study, interpreted the results and drafted the paper. RDS is guarantor of the study.

**Funding** This study was funded by the Department of Health in England Policy Research Programme (Policy Research Unit in Behaviour and Health (PR-UN-0409-10109)). LC is funded by an MRC Fellowship Grant (MR/P021999/1).

**Disclaimer** RDS affirms that the manuscript is an honest, accurate and transparent account of the study being reported, that no important aspects of the study have been omitted and that any discrepancies from the study as planned (and, if relevant, registered) have been explained. Representatives of the Department of Health had no role in the data collection, analysis or interpretation and no role in the study design or in writing the manuscript. The views expressed in this paper are those of the authors and not necessarily those of the Department of Health in England.

**Competing interests** None declared.

**Patient consent** Not required.

**Provenance and peer review** Not commissioned; externally peer reviewed.

**Data sharing statement** The data for this study were purchased from Kantar Worldpanel but its use is restricted to the persons named in the purchase contract which forbids the users to share the data with other potential (unnamed on the contract) users. Data access requests should be directed to Kantar Worldpanel.

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
