## [Reviewer comments · BMJ Open]

ARTICLE DETAILS

TITLE (PROVISIONAL)	Are sweet snacks more sensitive to price increases than sugar-sweetened beverages: analysis of British food purchase data
AUTHORS	Smith, Richard; Cornelsen, Laura; Quirmbach, Diana; Jebb, Susan; Marteau, Theresa

VERSION 1 – REVIEW

REVIEWER	Miranda Blake Associate Research Fellow, Deakin University, Australia I am also in the process of publishing papers on price elasticities and responses to pricing policies on SSBs (but using a different method and not comparing snack and beverage categories).
REVIEW RETURNED	03-Oct-2017

GENERAL COMMENTS	This paper advances the research on a topical issue and was an interesting read. Parts of the paper could be made clearer for a non-economist audience (both in body and appendix). The paper would also benefit from an additional proof read. Regarding presentation of results, I have added further comments in the attached but some reformatting of the figures and tables would be helpful. Overall comments This paper advances the research on a topical issue and was an interesting read. Parts of the paper could be made clearer for a non-economist audience (both in body and appendix). The paper would also benefit from an additional proof read. Introduction Nice introduction of currency of topic with clear rationale. Methods p. 4 Line 36. Would be good to begin this section with an overview of the methods used, including defining price elasticities. p.4 line 52-4. It would be good here or elsewhere, particularly in discussion to expand on why you are interested in different income groups. Is it because of differential purchasing of sugary foods? Equity concerns? p. 5 Lines 7-18. How did you decide these categories? These categories make sense from a dietary pattern perspective. Do they have similar sugar contents per category? Is there evidence that price elasticities are similar within each category? p.5, lines 36-7. Are these income tertiles? How were the ranges chosen? p.7 line 3, delete typo “cakes” in- “...significant reduction cakes, and for cakes....”
--

p.7 line 7-14. Would be good to get a sense of the magnitude of difference between income groups (where there are differences), since the graphs are separate for income groups.

Results

Text is well written. See comments on tables below.

Discussion

p.7 line 30. Insert "was" after "confectionary".

p.7 lines 30-45. This is an interesting discussion of the findings and implications. It would also be interesting to consider the sugar content of the food categories, and more on the proposed pricing structure. Currently, the UK levy is not a % price increase per se, but rather a specified absolute price increase for different sugar contents. If the same levy was applied to snack foods or other categories (e.g. per 100g rather than per 100mL), would this be a similar % price increase across categories? This is relevant, because currently you are comparing price elasticities (% price change), but the actual change in price (and therefore changes in purchase) may differ across categories.

p.7 line 53. Change typo "bought" to "brought".

p.7 lines 53-55. What is the basis for this assumption? In general, do you know what proportion of sweet snacks or SSBs are purchased at the supermarket in the UK? (Price sensitivity may differ by purchasing context).

Conclusion

p.8 Line 20-21. "especially in children". Please clarify- do you mean that the priority is reducing sugar intake in children, not that children's purchases are particularly susceptible to pricing changes?

Table 1

Lines 11-18. Indicate that these are mean (SD).

Lines 21-25. Round all to 1 decimal place.

Include all acronyms in footnote (HH, GCSE)

Table 2

p.12 line 25 'Rest food and drink' accounts for nearly half of sugar purchased. There are several subcategories that are significant sources of added sugar that might be of interest to pull out here, and to calculate separate price elasticities for. For example 'Breakfast cereals' and 'Condiments'. I'm unsure of the appropriateness of including 'dairy & eggs' and 'fruit and vegetables' in this table, since the sugar content is unlikely to be added sugar (are you proposing taxing items based on total or added sugar?). Could you also please add in the footnote here that flavoured milks are included in 'other soft drinks' rather than 'diary and eggs'.

Table 3

Lines 11-28. These are proportions, not percentages as indicated by the key in the first column.

Line 29. Does 'unit price' refer to price per item or price per serve?

Figures 1-4

Some of the food categories differ across the different figures. E.g. for 'price of sugary soft drinks' Fig 1 shows 'sugary soft drinks', 'other soft drinks' and 'cake-type snacks', but figure 2 shows and 'fruit and veg'. Please update figures for consistency or explain rationale for differing categories.

Appendix I

p.22 line 11. It would be helpful to readers who may not be familiar with demand models to define all parameters (α , β etc).

p. 22 Line 28 and p.23 line 17- amend errors around citation.

Appendix II

No comments, this is quite clear.

REVIEWER	Carlos Guerrero National Institute of Public Health, Mexico
REVIEW RETURNED	15-Oct-2017

GENERAL COMMENTS	The paper addresses an important topic related to public policy. However I would suggest some issues: There are more fiscal policies on SSB around the world. I would suggest to mention more. Maybe it would be necessary to discuss further the price elasticities that were estimated, comparing with other estimations made around the world, and mentioning the differences by method and type of data, e.g. cross section versus aggregated time series data. Also, I think it would be important to discuss the implications of the estimates with regard to public policies, e.g. changes in demand for such products or tax collection. I would suggest to use the term "Price-elasticity" instead of "price sensitivity".
---

REVIEWER	Gordon H. Guyatt Department of Health Research Methods, Innovation, and Impact McMaster University Canada
REVIEW RETURNED	30-Oct-2017

GENERAL COMMENTS	The statistical methods used Smith and colleagues seem to be common in the economic literature. A few points of consideration are outlined below:  1) Given that most readers of BMJ Open are not familiar with economic models, the authors should present a description of the Almost Ideal Demand System intelligible to BMJ Open readers in the introduction or the methods section. 2) In the discussion section, the authors should acknowledge the indirectness of the evidence addressing their research question. The authors use data on variations in the price for various products to estimate the impact of taxation of sugary foods. However, the price variations used in the analysis are not due to taxation. It is possible that the impact of taxation on demand for sugary foods may be different than the impact of regular price variations. For example, the public might actively oppose increased taxation by more ardently avoiding foods that are taxed. 3) Of the approximately 36,000 households surveyed, 4,075 ($\approx 11\%$) were excluded for missing information on sociodemographic variables. The impact of this on the representativeness of the sample should be acknowledged. Total amount of sugar purchased per day per person and purchases of sugar by food group can be compared between the analytic sample (32,249) and 4,075 households excluded from analysis as a way of gauging whether notable differences exist between the two groups. 4) The relationship between price and demand for food products may follow a non-linear relationship. The authors should describe steps they took to explore potential non-linearity and justify their final decision to use a linear model.
--

	5) There are no measures of variability presented on figures or in the main results in Appendix 2. Presenting confidence intervals could help readers better interpret the range in which price elasticities are likely to fall. 6) The authors present four separate models (full sample, low income, medium income, and high income) and consider price elasticities of various food categories and combinations thereof. In the figures, the authors present the statistically significant elasticities. Given the number of statistical tests performed, the authors should use a more stringent p-value than 0.05 – no higher than 0.01 and perhaps even lower.
--	--

VERSION 1 – AUTHOR RESPONSE

Reviewer 1

Overall comments

1) This paper advances the research on a topical issue and was an interesting read. Parts of the paper could be made clearer for a non-economist audience (both in body and appendix). The paper would also benefit from an additional proof read.

Thank you! We have made several edits to improve readability for a non-economist audience (see also below) and have carefully proof read the paper.

Methods

2) p. 4 Line 36. Would be good to begin this section with an overview of the methods used, including defining price elasticities.

We have added more detail to the beginning of the 1st paragraph in Methods section and define elasticities

3) p.4 line 52-4. It would be good here or elsewhere, particularly in discussion to expand on why you are interested in different income groups. Is it because of differential purchasing of sugary foods? Equity concerns?

We look at income groups primarily to allow possible differences in behaviours. We have specified this in the methods section.

4) p. 5 Lines 7-18. How did you decide these categories? These categories make sense from a dietary pattern perspective. Do they have similar sugar contents per category? Is there evidence that price elasticities are similar within each category?

During the revision process and particularly in answering this question, we decided to re-classify the non-alcoholic drinks according to the proposed levy in the UK for sugary drinks producers (from April 2018). We believe this provides a sound justification as well as relevancy as a number of countries have proposed measures similar to the UK where the levy is multi-tiered based on the sugar content of the beverages (e.g. Catalonia in Spain, Thailand and proposed levies in Estonia, Ireland, South-Africa).

In the proposed structure for the levy, drinks are categorised into three groups (with >8g of sugar/100ml; between 5-8g of sugar/100ml and with no added sugar and other excluded beverages (e.g. milk-based etc.)). The levy will apply to the first two categories. We have thus created 4 categories for non-alcoholic beverages (high-sugar, medium-sugar, low-sugar and other beverages (milk-based, fruit juice with no added sugars and water).

Sweet snack foods are in three separate categories based on product similarity (e.g. biscuits, candy & chocolate, cakes). We are not aware of studies estimating demand for such highly disaggregated food groups so cannot comment on whether a priori we would expect different type of biscuits for example to have similar elasticity.

Remaining foods are in five categories rather than one to allow for greater balance in the expenditure shares across food groups to ease estimation.

5) p.5, lines 36-7. Are these income tertiles? How were the ranges chosen?

These are based on the categories provided in the data (8 categories each by £10,000 up to £70,000). As we only hold data by categories we could not create tertiles.

6) p.7 line 3, delete typo “cakes” in- “...significant reduction cakes, and for cakes....”

Corrected

7) p.7 line 7-14. Would be good to get a sense of the magnitude of difference between income groups (where there are differences), since the graphs are separate for income groups.

Our analysis plan did not include testing for differences; however, we have added more detail to results section, pointing out more magnitudes of the changes in demand if the prices for different foods and beverages increases.

Results

Text is well written. See comments on tables below.

Discussion

8) p.7 line 30. Insert “was” after “confectionary”.

Corrected

9) p.7 lines 30-45. This is an interesting discussion of the findings and implications. It would also be interesting to consider the sugar content of the food categories, and more on the proposed pricing structure. Currently, the UK levy is not a % price increase per se, but rather a specified absolute price increase for different sugar contents. If the same levy was applied to snack foods or other categories (e.g. per 100g rather than per 100mL), would this be a similar % price increase across categories? This is relevant, because currently you are comparing price elasticities (% price change), but the actual change in price (and therefore changes in purchase) may differ across categories.

We have added to Table 2 the average sugar content across the food groups which clearly indicates, as expected, that in comparison to beverages the sugar content in sweet snacks is considerably higher (comparing 100g vs 100ml) though varies somewhat between the three groups we use. We have added the above point to the discussion but want to avoid speculation on any possible pricing structure so we refrain from providing any concrete figures how much prices are likely to change.

10) line 53. Change typo “bought” to “brought”.

Corrected

11) p.7 lines 53-55. What is the basis for this assumption? In general, do you know what proportion of sweet snacks or SSBs are purchased at the supermarket in the UK? (Price sensitivity may differ by purchasing context).

We have this information only on the volume of sales of soft drinks (~15% of the total volume is bought on-licenced trade). However, as we don't have this information by income groups or for the sweet snacks so we have removed this sentence altogether.

Conclusion

12) p.8 Line 20-21. "especially in children". Please clarify- do you mean that the priority is reducing sugar intake in children, not that children's purchases are particularly susceptible to pricing changes?

We mean the former. We have edited the sentence, which should be clearer now.

13) Table 1

Lines 11-18. Indicate that these are mean (SD).

Lines 21-25. Round all to 1 decimal place.

Include all acronyms in footnote (HH, GCSE)

Thanks, all the above are now corrected.

14) Table 2 p.12 line 25 'Rest food and drink' accounts for nearly half of sugar purchased. There are several subcategories that are significant sources of added sugar that might be of interest to pull out here, and to calculate separate price elasticities for. For example 'Breakfast cereals' and 'Condiments'. I'm unsure of the appropriateness of including 'dairy & eggs' and 'fruit and vegetables' in this table, since the sugar content is unlikely to be added sugar (are you proposing taxing items based on total or added sugar?). Could you also please add in the footnote here that flavoured milks are included in 'other soft drinks' rather than 'dairy and eggs'.

Our primary interest in this work was to understand how price changes of sweet snacks and SSBs influence the demand for either, rather than understanding what is the category that makes most sense to tax (which would require somewhat different analytical approach). The focus on SSBs and sweet snacks arises from a) the proposed levy that is about to come to force in spring 2018 in the UK and b) sweet snacks can be eaten on the go, without the need for utensils or further preparation and thus providing an "easy" substitute to replace possible reduction in sugar from SSBs and c) it is a group for which advocates are already calling similar measures to the sugary drinks levy to be implemented. We believe that further separating the remaining food groups to pull out specific groups that may contain relatively higher levels of sugar would distract from our main focus and is therefore not included here.

The reason we have separated out fresh foods (e.g. meat fish, dairy & eggs, fruits and vegetables) is pragmatic. If we combined these with rest of the foods, we would have one very large category (accounting for approximately 70% of the total food and beverage expenditure). Having eight categories with very small expenditure shares (most around 2-3%) and one large (70%) complicates estimation procedures.

If the reviewer believes it is less confusing we can remove food groups beyond sweet snacks and beverages in Table 2?

We have added a note to tables that milk-based drinks are with other soft drink category rather than dairy & eggs.

15) Table 3. Lines 11-28. These are proportions, not percentages as indicated by the key in the first column.

Corrected

Line 29. Does 'unit price' refer to price per item or price per serve?

With the exception of cakes and chocolate & confectionery, the unit price refers to price per L or Kg. Volume for cakes and chocolate & confectionery is measured in items, which we have specified but cannot convert into Kg.

16) Figures 1-4. Some of the food categories differ across the different figures. E.g. for 'price of sugary soft drinks' Fig1 shows 'sugary soft drinks', 'other soft drinks' and 'cake-type snacks', but figure 2 shows and 'fruit and veg'. Please update figures for consistency or explain rationale for differing categories.

The figures show the relationships we found to have significance (in the revised estimates based on 95% confidence intervals not including a zero). We have clarified this in the figure notes and refer to the appendix for the full set of estimates.

17) Appendix I. p.22 line 11. It would be helpful to readers who may not be familiar with demand models to define all parameters (α , β etc).

We have now added this detail ensuring all parameters are defined

18) p. 22 Line 28 and p.23 line 17- amend errors around citation. These are corrected now

Reviewer: 2

1) There are more fiscal policies on SSB around the world. I would suggest to mention more.

The reviewer is right and we have expanded the introduction to include more countries that have introduced this measure.

2) Maybe it would be necessary to discuss further the price elasticities that were estimated, comparing with other estimations made around the world, and mentioning the differences by method and type of data, e.g. cross section versus aggregated time series data. Also, i think it would be important to discuss the implications of the estimates with regard to public policies, e.g. changes in demand for such products or tax collection.

We have now added to the discussion comparing our estimates with those from meta-analyses. We have used the latter because there are too many individual papers to draw comparisons upon.

3) I would suggest to use the term "Price-elasticity" instead of "price sensitivity".

We have used "sensitivity" to avoid economic jargon given the main reader of the journal is not trained in economics. However we have within the text where possible used more of the term of price elasticity.

Reviewer: 3

The statistical methods used Smith and colleagues seem to be common in the economic literature. A few points of consideration are outlined below:

1) Given that most readers of BMJ Open are not familiar with economic models, the authors should present a description of the Almost Ideal Demand System intelligible to BMJ Open readers in the introduction or the methods section.

We have added more detail to the methods section that should be approachable to readers without familiarity of the demand models.

2) In the discussion section, the authors should acknowledge the indirectness of the evidence addressing their research question.

The authors use data on variations in the price for various products to estimate the impact of taxation of sugary foods. However, the price variations used in the analysis are not due to taxation. It is possible that the impact of taxation on demand for sugary foods may be different than the impact of regular price variations. For example, the public might actively oppose increased taxation by more ardently avoiding foods that are taxed.

We agree and have added this now to the discussion

3) Of the approximately 36,000 households surveyed, 4,075 ($\approx 11\%$) were excluded for missing information on sociodemographic variables. The impact of this on the representativeness of the sample should be acknowledged. Total amount of sugar purchased per day per person and purchases of sugar by food group can be compared between the analytic sample (32,249) and 4,075 households excluded from analysis as a way of gauging whether notable differences exist between the two groups.

The average sugar purchases are estimated based on weighted data to population level so the full sample in table 2 actually refers to the sample of 32620 households (all households that appear in the data in 2013). We apologise we did not make that clearer. We do this to avoid any errors, as we do not have information on whether grossing weights would need to be adjusted when aggregating sample to population level when excluding households that have not reported incomes. We have clarified this in the notes of the table.

We did however calculate sugar purchases excluding households with missing income as the reviewer suggests to check and the averages are very similar, particularly in the less aggregated first 9 groups (see below the average difference is by 1g). Thus, we do not believe excluding these households have influenced our conclusions we draw from the study.

Sugar purchased per day/person in 2013 (n=326,20) Excluding households where income is missing in 2013 (n=29,066)

Purchases of sugar by food group (g)	123.2	108.8
High-sugar soft drinks	6.3	5.7
Mid-sugar soft drinks	0.6	0.5
Low-sugar soft drinks	1.1	0.9
Other soft drinks (milk-based drinks, water, fruit juice)	3.9	3.5
Alcohol	2.0	1.8
Cookies (incl cereal fruit bars)	7.1	6.2
Chocolate & confectionary	7.7	6.8
Cake-type snacks	2.2	1.9
Savoury snacks	0.6	0.5
Fresh & frozen unprocessed meat, fish	0.5	0.5
Dairy & eggs	15.9	13.9
Fruit & Vegetables	17.6	15.5
Rest food & drink	57.8	51.0
Average difference	1.1	

4) There are no measures of variability presented on figures or in the main results in Appendix 2. Presenting confidence intervals could help readers better interpret the range in which price elasticities are likely to fall.

We have included confidence intervals to elasticities tables in Appendix 2 as well as the figures.

These are based on bootstrap standard errors due to possible bias arising in conventional standard errors from the 2-step estimation. In fact we have adjusted our interpretation of the findings also to be based on confidence intervals rather than p-values.

5) The authors present four separate models (full sample, low income, medium income, and high income) and consider price elasticities of various food categories and combinations thereof. In the figures, the authors present the statistically significant elasticities. Given the number of statistical tests performed, the authors should use a more stringent p-value than 0.05 – no higher than 0.01 and perhaps even lower.

We have now shown the 95% confidence intervals based on bootstrap standard errors (bias corrected) and discuss estimates with respect to this parameter rather than the p-value

Please note additionally that in the process of revision we discovered an error in the code relating to the units of savoury snacks (these were initially coded as measured by “item” while these should have been in kg). This has now been corrected.

We had also reported the wrong number of observations for the full sample, which is now corrected (originally, we had reported observations including households with missing income data).

VERSION 2 – REVIEW

REVIEWER	Miranda Blake Deakin University, Global Obesity Centre, Australia
REVIEW RETURNED	22-Dec-2017

GENERAL COMMENTS	Overall The authors have responded appropriately to comments. The paper is now easier to read for a non-economist audience while still informing a complex policy issue. There were a few small additional comments. Minor suggestions Since you frequently refer in the text to “SSB” and “sweet snack” sugar content/ expenditure etc in the results, but these categories are disaggregated in Tables 2 and 3, it would make it easier to look from the text to the tables if you added these two aggregated categories in Tables 2 and 3. The inclusion of the specific % change in purchases in your text explanations of Figures 2 to 4 are helpful. For consistency and to introduce the concept earlier, I suggest doing the same for the text relating to Figure 1. Discussion- It would be good to acknowledge that you have tested the impact of varying the prices of target foods and beverages in isolation (e.g. high-sugar soft drinks), rather than whole groups that might be taxed under a policy scenario (e.g. high-sugar soft drinks plus medium-sugar soft drinks). Therefore demand responses to policy may vary. TABLE 2- I am satisfied with the explanation for retaining the food groups as given with the manufacturer reported sugar content. Perhaps just indicate in methods and/or footnote of Table 2 that “sugar” refers to total sugars as reported by the manufacturer for
--

	clarity. Minor grammatical corrections p.5 line 29. From Table 2, should average amount of sugar intake from sweet snacks be 17.1g? (biscuits + chocolate + cakes) p.6, line 30-31 Suggest inserting the amount of sugar from SSBs into the text here- slightly confusing as previously sentence refers to sugar from sweet snacks and ALL beverages. p.6 line 41. Suggest change “sugary soft drinks” to “high and medium sugar soft drinks” for clarity p.7 line 30 replace “sugary drinks” with “high-sugar soft drinks” p.8 line 42 Insert “to” between “likely” and “change”
--	--

REVIEWER	Carlos Manuel Guerrero López National Institute of Public Health, Mexico
REVIEW RETURNED	11-Jan-2018

GENERAL COMMENTS	The paper aims to answer an important research question, which is quite relevant for policy purposes. However, some minor issues should be addressed:  1. In the abstract, the outcome of interest is called "sensitivity of food and beverage purchase to changes in their own price". However, in the methods, that is called price elasticity. I would suggest to use the same term of price elasticity in the abstract. 2. In line 45, page 4, authors mention that expenditure is a proxy for income. Could you further explain this? I.e. correlation between expenditure and income, the ratio income/expenditure, or how these varies according to income level? 3. In table 2 or 3, I would suggest to include a column with the proportion of zeros in the purchases for the different products and discuss how this can affect the estimates or interpretations. In appendix 1, the proportion for some products is mentioned but it is worthwhile to mention the proportions for all products. 4. In Discussion, in last paragraph of page 8, I would suggest to include another paper with similar methods and outcomes. I attach the PDF for the paper. 5. Although it is implicit, I suggest that conclusions should be bounded to the British context, since in some other countries, the proportion of sugars from SSB in all sugar intake represents nearly 70% (Mexico case, for instance). In those cases, public health benefits could be larger if tax are on SSBs.
---

REVIEWER	Gordon H. Guyatt McMaster University Canada
REVIEW RETURNED	20-Dec-2017

GENERAL COMMENTS	The manuscript by Smith and colleagues has been much improved by the addition of confidence intervals to the figures and tables, as well as the interpretation of results based on confidence intervals rather than statistical significance.
---

	A few additional points of consideration are outlined below: 1) P. 4, Lines 40-41 The definition provided for price elasticities could be made more clear. Consider replacing with the following: The impact, or sensitivity, of demand for a product to price changes is termed the price elasticity of demand. 2) P.6, Lines 3-11 It seems that the role of the funding source appears in the wrong section of the article. It appears to have been duplicated from page 3. 3) P. 6, Lines 40-44 The numbers presented in Table 3 do not represent what is reported in this paragraph. Please clarify where these numbers come from. 4) The Living Cost and Food Survey is referenced at numerous points in the article as evidence of the representativeness of the Kantar Worldpanel database (e.g., P.6, Lines 15-20; P. 9, Lines 10=14) . Providing a brief description of the survey may improve the clarity of the manuscript. 5) P. 7, Line 7 Period is missing at the end of the sentence. 6) P.8, Line 40-44 Add 'to' between 'likely' and 'change'.
--	--

VERSION 2 – AUTHOR RESPONSE

Reviewer: 3

A few additional points of consideration are outlined below:

1) P. 4, Lines 40-41

The definition provided for price elasticities could be made more clear. Consider replacing with the following: The impact, or sensitivity, of demand for a product to price changes is termed the price elasticity of demand.

*Thank you for pointing this out – indeed it was not well phrased. We have clarified this now as suggested (1st sentence in Methods).

2) P.6, Lines 3-11

It seems that the role of the funding source appears in the wrong section of the article. It appears to have been duplicated from page 3.

*Thank you for pointing this out. We have removed this from page 3.

3) P. 6, Lines 40-44

*The numbers presented in Table 3 do not represent what is reported in this paragraph. Please clarify where these numbers come from.

These numbers were calculations based on the numbers shown in table 3. We have added additional rows which hopefully clarify this (another reviewer had a suggestion along these lines as well).

4) The Living Cost and Food Survey is referenced at numerous points in the article as evidence of the representativeness of the Kantar Worldpanel database (e.g., P.6, Lines 15-20; P. 9, Lines 10=14) . Providing a brief description of the survey may improve the clarity of the manuscript.

*We have added an additional phrase “LCF is a survey of household spending and the cost of living in the UK reflecting household budgets and is conducted by the UK Office for National Statistics” (footnote 2).

5) P. 7, Line 7

Period is missing at the end of the sentence.

*Corrected

6) P.8, Line 40-44

Add ‘to’ between ‘likely’ and ‘change’.

* Corrected

Reviewer: 1

There were a few small additional comments.

1) Since you frequently refer in the text to “SSB” and “sweet snack” sugar content/ expenditure etc in the results, but these categories are disaggregated in Tables 2 and 3, it would make it easier to look from the text to the tables if you added these two aggregated categories in Tables 2 and 3.

*We have added a column specifying categories for SSBs and sweet snacks for easier view of the tables. As pointed out by other reviewer we also added few additional rows with average statistics.

2) The inclusion of the specific % change in purchases in your text explanations of Figures 2 to 4 are helpful. For consistency and to introduce the concept earlier, I suggest doing the same for the text relating to Figure 1.

*We have added these explanations now.

3) Discussion- It would be good to acknowledge that you have tested the impact of varying the prices of target foods and beverages in isolation (e.g. high-sugar soft drinks), rather than whole groups that might be taxed under a policy scenario (e.g. high-sugar soft drinks plus medium-sugar soft drinks). Therefore demand responses to policy may vary.

*We agree and have added this now to the discussion under limitations

4) TABLE 2- I am satisfied with the explanation for retaining the food groups as given with the manufacturer reported sugar content. Perhaps just indicate in methods and/or footnote of Table 2 that “sugar” refers to total sugars as reported by the manufacturer for clarity.

*We have clarified this now in the notes of table 2.

5) Minor grammatical corrections

p.5 line 29. From Table 2, should average amount of sugar intake from sweet snacks be 17.1g? (biscuits + chocolate + cakes)

*Yes that is the case and thanks for pointing this out! We have corrected this now.

6) p.6, line 30-31 Suggest inserting the amount of sugar from SSBs into the text here- slightly confusing as previously sentence refers to sugar from sweet snacks and ALL beverages.

*We have added the sugar content from SSBs to the following sentence to avoid duplicating as the following sentence already half-way provided this information.

7)

- p.6 line 41. Suggest change “sugary soft drinks” to “high and medium sugar soft drinks” for clarity
- p.7 line 30 replace “sugary drinks” with “high-sugar soft drinks”
- p.8 line 42 Insert “to” between “likely” and “change”

*We have done the above three changes as suggested

Reviewer: 2

1. In the abstract, the outcome of interest is called "sensitivity of food and beverage purchase to changes in their own price". However, in the methods, that is called price elasticity. I would suggest to use the same term of price elasticity in the abstract.

* We have rephrased this (to ensure that the definition of elasticity is still clear).

“Change in food and beverage purchases due to changes in their own price and the price of other foods or beverages measured as price elasticity of demand for the full sample and by income groups.”

2. In line 45, page 4, authors mention that expenditure is a proxy for income. Could you further explain this? I.e. correlation between expenditure and income, the ratio income/expenditure, or how these varies according to income level?

*We have removed this mention as indeed the usual reporting refers to total expenditure. As expenditure shares make up the total expenditure leading to issues of endogeneity, using income instead would be preferable but our wording was misleading and as we don't have income as continuous variable, it becomes irrelevant.

3. In table 2 or 3, I would suggest to include a column with the proportion of zeros in the purchases for the different products and discuss how this can affect the estimates or interpretations. In appendix 1, the proportion for some products is mentioned but it is worthwhile to mention the proportions for all products.

*Thanks for suggesting this. We added this to table 2 and a paragraph on it to results as it's an interesting indication of purchase frequencies/regularity as well.

4. In Discussion, in last paragraph of page 8, I would suggest to include another paper with similar methods and outcomes. I attach the PDF for the paper.

*Thanks for pointing this out. We have added this and another paper from Chile.

5. Although it is implicit, I suggest that conclusions should be bounded to the British context, since in some other countries, the proportion of sugars from SSB in all sugar intake represents nearly 70% (Mexico case, for instance). In those cases, public health benefits could be larger if tax are on SSBs.

*That is a good point and we have added more explicitly the reference to UK policy.